# Diagnostic and Therapeutic Utility of Extracellular Vesicles in Ocular Disease

**DOI:** 10.3390/ijms26020836

**Published:** 2025-01-20

**Authors:** Vladimir Khristov, Sarah R. Weber, Mireille Caton-Darby, Gregory Campbell, Jeffrey M. Sundstrom

**Affiliations:** 1Penn State Hershey College of Medicine, Pennsylvania State University, Hershey, PA 17033, USA; vkhristov@pennstatehealth.psu.edu (V.K.); gcampbell4@pennstatehealth.psu.edu (G.C.); 2Department of Ophthalmology, Penn State University, Hershey, PA 17033, USA; sweber2@pennstatehealth.psu.edu (S.R.W.); mcatondarby@pennstatehealth.psu.edu (M.C.-D.)

**Keywords:** extracellular vesicles, biomarkers, therapeutic, diagnosis, vitreous humor, tear film

## Abstract

Extracellular vesicles (EVs) are lipid bilayer particles released by virtually all cells, with prominent roles in both physiological and pathological processes. The size, number, and molecular composition of released EVs correlate to the cells of origin, modulated by the cell’s environment and pathologic state. The proteins, DNA, RNA, and protein cargo carried by EVs are protected by degradation, with a prominent role in targeted intercellular signaling. These properties make EVs salient targets as both carriers of biomarkers and potential therapeutic delivery vehicles. The majority of EV research has focused on blood, urine, saliva, and cerebrospinal fluid due to easy accessibility. EVs have also been identified and studied in all ocular biofluids, including the vitreous humor, the aqueous humor, and the tear film, and the study of EVs in ocular disease is a new, promising, and underexplored direction with unique challenges and considerations. This review covers recent advances in the diagnostic and therapeutic use of ocular EVs, with a focus on human applications and key preceding in vitro and in vivo animal studies. We also discuss future directions based on the study of EVs in other organ systems and disease sates.

## 1. Introduction

Extracellular vesicles (EVs) are lipid bilayer particles released by virtually all cells, and their contents are shown to directly correlate to the cells of origin [1]. Initially thought to be a waste-disposal mechanism, their functions are now known to include intercellular communication, disposal of undesirable material, and transfer of functional proteins and ribonucleic acid (RNA) [1]. In fact, the EV lumen has been shown to contain a variety of functional proteins, messenger RNA (mRNA), transfer RNA (tRNA), ribosomal RNA (rRNA), micro RNA (miRNA), and double-stranded DNA (dsDNA) fragments [2]. The definition of EV subclassification is an ongoing discussion. Broadly speaking, groupings are evolving from a descriptive classification based on size towards a classification based on the mechanism of origin. Exosomes are generally smaller than 100 nm in diameter and produced via endosome maturation into multivesicular bodies (MVBs), whereas ectosomes are larger 100–1000 nm particles formed when surface blebs are split off from the plasma membrane [3,4]. Furthermore, cells undergoing apoptosis release large apoptotic bodies through plasma membrane blebbing that are 1–5 µm in diameter [5]. Regardless of their origin, EVs stabilize their molecular cargo, delaying degradation and making them appealing targets in both diagnostic and therapeutic applications [6,7].

### 1.1. EV Isolation

EVs have been isolated from a variety of biofluids, including blood, urine, and CSF [8,9,10]. Interestingly, EVs are capable of crossing cellular barriers while retaining their cargo, implying that biofluids not proximal to the cells of origin can still yield useful information [11]. However, the biofluid most proximal to the disease process should provide the greatest differences between diseased and healthy EVs due to the higher relative concentration of pathologic EVs in disease-proximal biofluids [6]. A variety of methods can be used to isolate EVs from biofluids, including precipitation, ultracentrifugation, size-exclusion chromatography, microfluidic methods, and affinity purification based on membrane lipids and surface proteins [12,13,14]. The choice of isolation method depends on the biofluid source, the anticipated downstream analysis, and the relative concentration of the EVs of interest. For example, affinity purification may be useful to isolate tissue-specific EVs that make up a small proportion of all EVs in a biofluid, thus increasing the potential signal-to-noise ratio at the expense of EV concentration [15,16]. In contrast, studying the total EV population in a biofluid captures both the EVs derived from pathologic tissues and from the body’s response to the underlying pathology. For example, affinity purification may be useful for the discovery of novel disease-associated EV biomarkers but may lead to undesirable loss of EVs when detecting a highly specific disease-associated biomarker.

The large variety of EV isolation methods and biotech companies competing for market share have resulted in remarkably little standardization across the field of EV research and a lack of unified consensus, with hundreds of isolation methods and protocols [17,18]. Attempts to establish the minimal requirements for the isolation and study of EVs in order to ensure validity, reproducibility, and translational potential are ongoing and should be adhered to [19,20,21].

### 1.2. EV Quantification and Analysis

The most common quantification of EVs is their concentration; however, their small size presents unique challenges. Measuring the protein concentration in the EV fraction of a biofluid is a crude way to assess the concentration, although this method is confounded by non-EV proteins and only a general correlation between EVs and protein concentration [22]. Furthermore, because each EV purification method can alter the proportion of non-EV proteins in the isolate, simply measuring the protein concentration is not a robust way to compare EVs isolated through different methods. Due to a substantial proportion of EVs being smaller than 200 nm, their analysis through light microscopy is generally limited by Abbe’s diffraction limit, which is thought to be possible with fluorescent labeling and image analysis [23]. Scanning and transmission electron microscopy can easily image EVs, albeit at the expense of time and sample preparation [24]. The most accessible and widely used method for assessing EV size, concentration, and surface markers at this time remains nanoparticle tracking analysis (NTA), which infers the hydrodynamic particle diameter from high resolution assessment of the EV’s Brownian motion in solution, coupled with its fluorescent particle labeling capability [25]. This method is also not without its pitfalls, as it will also count non-EV debris and clumped EVs within the target size range. This problem can be partially overcome via fluorescent labeling of EV surface markers, and novel technologies allowing for single EV capture, imaging, and characterization are being developed [26].

Assessment of proteomic EV contents has evolved from Western blotting and enzyme-linked immunosorbent assays towards advanced methods, such as single molecule localization microscopy [27]. Mass spectrometry analysis of EVs is another powerful tool to enable qualitative and quantitative analysis of protein, lipid, and metabolite cargo [28,29,30].

The genetic contents of EVs in biofluids can be interrogated with or without enrichment from the biofluid of interest. Next generation sequencing (NGS) was a major advancement in the study of EV genetic cargo, allowing for relatively fast assessment of mRNAs, small interfering RNA (siRNAs), and miRNAs, as well as small nucleotide polymorphisms [31].

### 1.3. Ocular EVs

The study of EVs in ocular disease is a new, promising, and underexplored direction, with tears, aqueous humor, and vitreous humor serving as potential EV-containing biofluids. EVs have been identified in all ocular biofluids, including the vitreous humor [30], the aqueous humor [32], and the tear film [33]. As shown with EVs derived from other tissues, the concentration and composition of ocular fluid EVs change with both the disease state and normal physiological processes. For example, Biasutto et al. showed that the retinal pigment epithelium (RPE) produced altered protein-laden EVs in response to oxidative stress [34]. Similarly, Demais et al. showed that Müller cells and neurons secrete exosomes with distinct molecular profiles and that the proteomic content of the EVs changes in response to ischemia [35]. Age-specific changes in vitreous EV miRNAs have also been observed in an animal model [36]. As shown with non-ocular tissues, the cellular origin of EVs can now be traced using novel techniques, such as multiplex proximity extension assays [37]. This review covers key studies in ocular EVs that have evolved into novel diagnostic and therapeutic applications (Figure 1).

## 2. Diagnostic Ocular Use of EVs

The use of ocular EVs for diagnosis is a rapidly developing field, although it is partially limited by the difficulty of collecting ocular fluids compared to other biofluids, such as blood, urine, and cerebrospinal fluid, as well as the limited collection volume. Nevertheless, collection of tear fluid and vitreous fluid from patients with infectious endophthalmitis and those undergoing vitrectomy has provided reliable samples for study. At this stage, only six clinical trials utilizing EVs for the diagnosis of ocular pathology have been identified on clinicaltrials.gov (Table 1). It is known that the concentration and size distribution of EVs are significantly different between several eye pathological processes, and changes in intraocular EV concentration following treatment, such as anti-vascular endothelial growth factor (anti-VEGF) injection, opens possibilities to study both disease and treatment responses [38]. Diagnostic ocular EVs have been studied in the setting of several pathological processes outlined below.

### 2.1. Infectious

Gandhi and Joseph showed that in patients with suspected fungal endophthalmitis, aquaporin-5 levels in EVs isolated from vitreous humor could be used to distinguish between culture-proven bacterial and fungal endophthalmitis [39]. Similarly, Rudraprasad et al. showed that complement cascade 8-alpha was elevated and calpain-8 was reduced in the vitreous-derived EVs of both culture-positive and culture-negative bacterial endophthalmitis patients compared to controls [40]. EV-based assays may therefore prove to be key diagnostic tests that guide the treatment of ocular infections.

### 2.2. Malignancy

Vitreous EVs have been studied in the context of ocular malignancy. Specifically, Pessuti et al. made the connection between aqueous humor, vitreous humor, and plasma samples in patients with uveal melanoma. They demonstrated that the EV concentration in the vitreous humor of uveal melanoma (UM) patients was higher than the concentration in the aqueous humor or the plasma, suggesting that a biofluid source proximal to the malignancy would provide the highest pathologic EV yield. Further mass spectroscopy analysis of uveal melanoma patient EVs showed a high percentage of shared protein cargo between plasma-, aqueous-humor-, and vitreous-humor (VH)-derived EVs, and the EVs carried proteins derived from melanocytes of the uveal tract [41]. Studies of miRNA in the EVs from uveal melanoma patients, in contrast, showed that UM VH samples contained a unique miRNA profile that only partially overlapped with corresponding plasma EVs, again suggesting VH as the optimal biofluid source [42]. Although ocular malignancy can be diagnosed clinically, analysis of ocular EVs can shed light on the pathophysiology of the disease, identify targetable tumor mutations, enable biomarker discovery, and allow for better disease monitoring.

### 2.3. Retinal Disease

EVs have also been investigated in the study of proliferative diabetic retinopathy (PDR), which is often a manifestation of systemic changes in the patient’s vasculature. Proteomic analysis of plasma- and vitreous-derived EVs of patients with PDR showed elevation in tumor-necrosis-factor-α-induced protein 8 (TNFAIP8) and confirmed its ability to pathologically alter human retinal microvascular endothelial cell migration and tube formation, indicating that TNFAIP8 can be used as a biomarker of PDR and potentially as a therapeutic target [43,44]. Recently, Shan et al. showed that while the size and concentration of VH EVs is not significantly different between diabetic and non-diabetic patients, the EVs of type II diabetics had the capacity to induce TNFα and IL1β inflammatory cytokine expression in macrophages [45]. Similarly, Wang et al. showed differential protein expression in the vitreous EVs of patients with PDR, including elevations in APOB, APOM, LDHA, and FCN3 [46]. Furthermore, lncRNA LOC100132249 has been implicated in endothelial dysfunction in PDR [47]. The characteristics of EVs have also been studied in other eye pathologies, including pathological myopia, suggesting that differentially expressed EV-miRNAs could predict the development of pathologic myopia maculopathy [48]. Further, pro-inflammatory vitreous EVs were elevated following retinal detachment, which could serve as prognostic factors for photoreceptor cell death and as potential therapeutic targets [49].

### 2.4. Corneal Disease

Tear fluid EVs (tEVs) in particular are appealing for both diagnostic and therapeutic purposes due to their easy accessibility and negligible risk to the patient compared to VH and aqueous humor (AH) sampling. Tear fluid contains abundant EVs sized 40–100 nm that can be subjected to further proteomic and genetic analysis [33,50,51]. Since the initial characterization of tEVs in 2016 by Grigor’eva et al., numerous diagnostic and therapeutic applications have been studied [33].

Several groups have reported the diagnostic potential of tEVs, as evidenced by distinct proteomic and genomic EV components in patients with ocular disease compared to healthy controls. Aqrawi et al. analyzed the EVs of patients with Sjogren’s syndrome using liquid chromatography-mass spectrometry (LC-MS), finding alterations in metabolism, protein folding, and the adaptive immune response [52]. Similarly, distinct proteome and RNA signatures in the tear EVs of patients with dry eye syndrome were found [53,54,55]. Tamkovich et al. showed differences in the miRNA expression in tEVs of patients with primary open angle glaucoma compared to controls [56]. In patients with thyroid eye disease, tEVs had distinct differences in cytokine levels, including Il-1 and IL-18, as well as C-reactive protein, MMP-9, and VCAM-1 [57,58]. Interestingly, even in non-inflammatory conditions, such as keratoconus, subtle differences in the makeup of tEVs was detected, indicating that studying their contents could shed light on the pathophysiology of this condition [59]. Even retinal diseases, such as age-related macular degeneration, diabetic macular edema, and diabetic retinopathy, were found to alter the miRNA and proteomic composition of tEVs [60,61]. Further studies are needed to elucidate the cellular source of tEVs and whether tEV changes in patients with retinal pathologies are a result of pathophysiology or treatment [59,62].

The study of tear EVs from patients undergoing treatment presents an exciting opportunity to not only learn about the disease mechanism but the response to treatment, as well. For example, through analysis with Olink proteomics, Thormann et al. showed distinct shifts from Th2/Th17 to Th1/Th17 cytokine profiles in patients with Dupilumab-associated ocular surface disease [63], which could be ameliorated through the application of bone marrow mesenchymal-stem-cell-derived exosomes [64]. Proteomic changes have also been detected in tEVs in the context of infection, such as herpes simplex virus [65], and they have been implicated in the spread of herpes simplex keratitis [66].

In summary, the use of EVs has a high potential to enhance diagnosis and the understanding of disease in the ocular setting (Figure 2). At this nascent stage, this field is in urgent need of safe and standardized biofluid collection methods, as well as analysis of EV characteristics, surface markers, and cargo. Additionally, EV isolation and analysis methods that are tailored to low volumes are urgently needed. The benefit of diagnostic sample collection must always be weighed against the risk of harm and the capacity to inform treatment and affect outcomes.

## 3. Ocular Therapeutic Use of EVs

In principle, several factors make EVs promising from a therapeutic standpoint, including protection of their molecular cargo from degradation, non-immunogenicity, and the promise of targeted delivery. Ocular therapeutic use of EVs derived from several cellular sources has also been thoroughly explored in animal models, with emerging applications in humans. The foundation of these studies is that through careful selection of the cell type and the environment, EVs with unique functional properties can be produced, with the goal of influencing pathologic cells and processes. The EV surface molecules, contents, and delivery vehicle can also be modified to target the EVs and alter their stability, transport, and uptake after delivery.

### 3.1. EV Manufacturing

Cells are the natural factories for producing EVs, and by selecting an appropriate cell type and cell culture conditions, EVs tailored for therapeutics can be produced [67]. A key idea distinguishing EV manufacturing from other therapeutics is that therapeutic EVs are a complex and heterogenous product, even before the introduction of any processing, modification, or storage steps. Their heterogeneity can be controlled only to a degree, and, ultimately, clinical application will depend on (1) safety and (2) efficacy.

Following the discovery that mesenchymal stem cells (MSCs) are therapeutically beneficial for a variety of diseases, and that their therapeutic effects can be mediated via EVs, MSC-derived EVs have been extensively studied in ocular and non-ocular applications [68,69,70]. Multiple advantages of MSC-EVs have emerged, including elimination of the risk of malignant transformation, enhanced biodistribution, simplified storage, and lower immunogenic potential [71,72,73]. It must be noted that the term MSC is broad, and it can refer to a variety of cell types and can therefore produce EVs with variable composition and function [74]. Nevertheless, in the majority of ocular therapeutic EV applications, the EVs are MSC-derived due to the large body of evidence, established current good manufacturing practices (cGMP) manufacturing, and clinical trials supporting MSC therapy.

Unfortunately, there is little consensus around creating a unified manufacturing protocol for MSC-EVs and, by extension, EVs derived from other cell types [75,76,77]. Following cGMP guidelines is crucial for regulatory approval [78]. Several considerations may influence the design of the manufacturing process, which may be unique to the clinical application. An autologous process can be more costly, with potentially lower immunogenicity, while an allogeneic process can be scalable, cheaper, more expedient, and allow for banking of a certified product [79]. Changes in the cell line, including cell age and passage number, may also be reflected in the resulting EVs, which could be addressed through monitoring of cellular age via the population doubling level and utilizing a tiered banking system [76,80]. The process of tiered banking has already been developed for autologous MSC therapy [81], and it addresses initial MSC heterogeneity and donor–donor variations. The creation of a validated master cell bank and subsequent working cell banks along with continuous quality control measures should be applied to EV manufacturing, as well.

The components of the cell culture media used for EV collection must be balanced for the optimization of cellular growth, EV production, and implications for downstream EV purification. Components, such as fetal bovine serum or human platelet lysate, are common in cell cultures but contain large amounts of EVs that would be purified along with EVs of interest [82]. Elimination of serum in cell culture media reduced the problem of contamination with serum EVs, although the resulting stress on EV-producing cells and the corresponding alteration in EVs’ composition are important considerations and may be beneficial [83,84].

Other cell culture parameters, such as cell seeding density [76] and hypoxia, can affect EV composition [85]. Through utilizing bioreactors rather than two-dimensional (2D) cell cultures, EV production volumes can be dramatically increased to achieve therapeutic amounts [86,87]. However, changes in the cellular environment also influence EV composition and are important considerations [88]. A bioreactor system also allows for continuous monitoring and optimization of cell culture conditions and scaling of manufacturing [89]. These changes are not only explained by larger cell density but also by decreased EV reuptake in a perfusion bioreactor system [90]. A focus on cells that can be grown in suspension culture will allow for easier scalability and control over culture conditions, which has been demonstrated with MSCs for the purposes of EV manufacturing for ocular use [91].

Regardless of the manufacturing method used, the next step in EV manufacturing involves the concentration and purification of the EVs from cell culture media. Multiple cGMP-compatible methods exist, among which precipitation, ultracentrifugation, and ultrafiltration are most scalable, although there is no consensus on the optimal method [92,93]. The scalability and complexity of the purification method are crucial considerations. While ultracentrifugation is applicable to small volumes of media, it is time-consuming and costly to scale. In contrast, a method such as tangential flow filtration can be utilized to scale up EV manufacturing by allowing continuous EV collection from the cell culture media in a bioreactor [94].

### 3.2. EV Modification

The same considerations that guide pharmacokinetics of small molecules and biologic drugs can be applied to EVs, whose inherent complexity and heterogeneity require a unique approach. The lipid bilayer of EVs is akin to the lipid bilayer of a cell characterized by proteins, proteoglycans, glycans, lectins, and lipids [95]. These surface molecules contribute to cell-specific targeting and internalization, while also presenting exciting opportunities to tailor these properties [96,97,98,99]. As an example, insertion of amphiphilic phosphatidylcholine into EVs significantly increased tumor cell internalization, while blocking caveolae-dependent endocytosis can reduce EV uptake [100,101]. Salunkhe et al. provide an excellent review of EV surface modifications for targeting purposes, and the same approaches can be applied to ocular therapeutic EVs [102]. For example, the rod, the cone, and the RPE can be targeted in age-related macular degeneration, as well as retinal ganglion cells in glaucoma and RPE cells in Stargardt disease.

The contents of EVs may be similarly modified for therapeutic effect. EVs can be loaded with adeno-associated viruses (AAVs), proteins, and small molecules either through endogenous treatment of EV donor cells or postproduction modification [103]. One application of EV loading was demonstrated by Li et al., who used sonication to load IL-10 into MSC-EVs, thereby extending its therapeutic effect in a mouse model of autoimmune uveitis [104]. An alternative approach of endogenous loading also explored by the same group utilized the transfection of MSCs with a lentivirus carrying IL-10-overexpressing plasmids, confirming its presence in the resulting MSC-EVs and its therapeutic effect in the mouse model, albeit without specifically measuring the IL-10 concentration in the EVs [105]. Other methods of exogenous cargo loading exist, including freeze–thawing, co-incubation, electroporation, sonication, and extrusion [106,107]. While EVs loaded via endogenous or exogenous loading methods demonstrate therapeutic benefits, exogenous methods may allow for more control over the manufacturing process. Nevertheless, the use of cell lines for endogenous protein production is extensively utilized in the biopharmaceutical industry and can inform this approach [108]. As MSCs are already in clinical trials, one approach could be to use the resulting GMP-grade medium for further EV isolation [109].

Loading of AAVs into EVs could be especially relevant to ocular applications given the recent drastic increase in the study of AAVs for retinal disease [110]. With multiple ongoing gene therapy trials for retinal diseases, EVs are poised to make incremental improvements through targeted delivery and mediation of the inflammatory response [111]. More studies are needed to evaluate whether EV encapsulation could enhance AAV delivery to deeper retinal layers and alter innate and adaptive immune responses [112].

One unresolved question key to clinical applications regarding EV loading and modification remains quality control, and robust assays are needed to characterize modification. A wide range of EV loading efficiencies has been reported depending on the encapsulation method and the cargo, generally ranging from 0.4 to 80% [113]. In the case of encapsulation of a therapeutic peptide for ocular use, Li et al. have shown that sonication is a method to both encapsulate IL-10 into EVs and measure the loading efficiency of 10.35% [104]. In the case of viral EV loading, assessment of the viral potency units should be used [114,115]. These results further reinforce the idea that EVs are a complex and heterogenous product by definition, and the introduction of cargo and modifications further increases their heterogeneity. Although this heterogeneity should be characterized and tracked, functional assessments of the loaded cargo and its ability to exert therapeutic effects in vitro and in vivo will remain perhaps the most important assay for regulatory approval after safety.

In summary, researchers aiming to develop ocular therapeutic EVs should aim to use manufacturing methods, modification methods, and assessments that will enable easy transition towards regulatory compliance. Current protocols to assess EV concentration, size distribution, and protein concentration should be continued at all points of the manufacturing process. Immortalization of cells through the use of oncogenes and upregulation of human telomerase reverse transcriptase should allow for large passage numbers of cell lines, but with the need to characterize a limit of population doubling. In the case of endogenous or exogenous cargo loading, researchers should focus on robust quality assurance protocols that characterize not only the presence of therapeutic cargo but also its function through in vitro and in vivo assays. These assays should be repeated at all stages of the manufacturing process to ensure that any modification processes and storage conditions do not impact efficacy. EV storage, stability, and potency over time are crucial considerations that need further study [116,117]. EVs release cargo during freeze–thaw cycles [118], and the introduction of cryopreservation reagents introduces added regulatory complexity and the possibility of contamination. Finally, and perhaps most importantly, the final EV product should be assessed from a safety standpoint, with a focus on pathogen contamination, such as mycoplasma, viruses, bacteria, endotoxins, and fungi. A summary of key considerations at each manufacturing step is outlined (Figure 3).

Adherence to these guidelines, along with cGMP principles, will pave the path towards regulatory compliance. Researchers should keep in mind that additional complexity, such as cell line modification, EV modification, cargo loading, EV encapsulation, and prolonged storage, all increase regulatory complexity and introduce the chance for errors. This complexity must always be weighed against the desired therapeutic benefit. It is not surprising that in initial ocular studies, unmodified EVs are derived from well-studied MSCs.

### 3.3. EV Dosing

It is crucial to address the question of EV dosing in order to accurately assess therapeutic efficacy, dynamics, and kinetics [119]. The difficulty in addressing this question lies in the natural complexity and heterogeneity of EVs, even if isolated in a controlled environment, such as a cell culture. The majority of existing pre-clinical studies utilized in vivo dosing based either on protein concentration or particle number, with large variability in the EV dose per kg of body weight [119]. While dosing based on protein concentration is a simple and quick method, it does not distinguish between EV protein, soluble proteins, and contaminants. In rodent models of ocular disease, there is heterogeneity in the units of measurement as well as the magnitude, with ranges from 1× to 3 × 10^9^ particles or 4.5 to 15 µg of protein delivered in 5 µL [120].

So far, doses of 10–50 µg in 50 µL have been used in human studies with minimal inflammatory reaction (Table 2). These amounts appear to be low relative to rodent studies, as the human eye has a volume of 6 mL compared to 0.15 mL for a rodent, suggesting that a human dose would need to be 40× larger (given that a rat eye has a volume of 0.15 mL and a human eye has a volume of ~6 mL), or at least 180 µg. A lower dose to achieve the same efficacy may be possible in the ocular setting through EV encapsulation.

Similarly, assessing EV size and concentration does not capture the identity and functionality of EV components and is not without variability [121]. Several promising approaches exist to overcome these challenges, including functional assays and quantification of effector molecules carried by the EVs [122,123,124], but these strategies have not yet been applied to human trials, and further development is urgently needed in this area.

Furthermore, through modification of the EV delivery medium, such as via encapsulation in a thermosensitive hydrogel by Tang et al., sustained EV release can be achieved in the ocular setting [125]. While having few ocular applications to date, encapsulation of microparticles, such as EVs, has been extensively studied in a variety of applications, including spinal cord injury [126], inflammatory bowel disease [127], and regenerative medicine [128,129]. There are multiple advantages to encapsulation, including mitigation of the immune response [130], further extending the stability of EVs’ structure and function. 3D printing [131,132] or microsphere encapsulation of EVs allows for spatiotemporal EV release in the target area [133,134,135]. For example, by modifying the hydrogel degradation properties, Li et al. demonstrated remarkable temporal control over the release of two functional populations of EVs from hydrogel-encapsulated microspheres [136]. Building on this knowledge, EV microsphere encapsulation has a high potential to synergize with other EV modifications and enable less frequent dosing in the ocular setting.

### 3.4. Intraocular EV Delivery

Numerous pre-clinical studies have been conducted to study the safety and distribution of intraocularly delivered EVs. Mathew et al. tested the fate of MSC-derived intravitreally administered EVs using in vivo and ex vivo rat models, showing that EVs were primarily endocytosed by the inner retina, peaking 14 days after injection and penetrating no deeper than the inner nuclear layer [137]. Attempts to alter the uptake and transport of EVs have also been investigated. Modification of the EV surface with cationic peptides enhanced exosome transport through the cornea and the vitreous humor [138]. Pollalis et al., in contrast, showed that retina-derived EVs penetrated both the inner and the outer nuclear layers, including inner plexiform layer, inner nuclear layer, outer plexiform layer, and outer nuclear layer. This is in contrast to MSC-EVs, which appear to be limited to the inner retina. Moreover, systemically administered EVs were also detected in the retina, suggesting that they can cross the blood–retina barrier [139]. Furthermore, modification of the EV surface with a targeting RGD peptide allowed the EVs to target cytomegalovirus sites in the retina [139].

Several other disease systems have demonstrated potential EV utility to counteract pathologic processes. For example, pro-angiogenic EVs can be produced from hypoxic-preconditioned endothelial cells to stimulate angiogenesis and facilitate nerve tissue repair [140], as well as recovery from focal brain ischemia in mice [141]. Similar mechanisms can be targeted to modulate ischemia-induced changes after retinal artery occlusion [142]. Mesenchymal-stem-cell derived EVs have been shown to reverse endothelial-to-mesenchymal cell transition (EMT) in endometrial repair and cancer pathogenesis [143,144,145]. Similarly, EMT is an important and targetable mechanism in retinal, lens, and corneal pathologic processes [146,147,148]. By utilizing the capacity of tissues to produce functional EVs in response to stressors, the potential exists to produce therapeutic EVs that could alter inflammatory, angiogenic, and proliferative pathways [149].

Indeed, multiple trials utilizing EVs are ongoing, which can be used as templates for potential EV application to ocular disease. For example, platelet-derived EVs have been shown to be a safe wound-healing treatment [150] beneficial for COVID-19-induced respiratory failure [151,152], and they may be part of the therapeutic mechanism behind neurological improvements in stroke patients treated with mesenchymal stem injections [153].

### 3.5. Ocular Surface Therapeutic EVs

The application of EVs to the surface of the cornea has been studied in the context of dry eye disease, recovery from corneal injury, and corneal infection. MSC-derived EVs have been shown to mitigate inflammation and restore homeostasis in a mouse model of dry eye disease [154,155]. EVs derived from mouse adipose mesenchymal stem cells could be used to promote diabetic corneal epithelial wound healing through activation of dendritic cells [156]. EVs derived from M2 macrophages or adipose tissue stem could be beneficial for treating dry eye disease by targeting ocular surface inflammation [157,158,159,160]. Furthermore, the EV surface can be modified to improve corneal epithelial recovery in dry eye disease [161]. Pathogen-produced EVs, such as those produced by Aspergillus fumigates, were found to alter immune cell function and increase the secretion of secretory immunoglobulin A in tear fluid, which could be utilized therapeutically to enhance the host response to fungal keratitis [162].

### 3.6. Intravitreal Therapeutic EVs

Therapeutic intravitreal EV application has largely centered on EVs derived from mesenchymal stem cells (MSCs) that have been extensively characterized in terms of their capacity for immune regulation and tissue regeneration [163]. Intravitreal injection of MSCs was found to be neuroprotective and mediated largely by EVs taken up by retinal neurons, ganglion cells, and microglia [164]. Therefore, MSC-derived EVs (MSC-EVs) are a promising cell-free therapeutic alternative to cell injection. The broad target of these interventions is the mitigation of an inflammatory microenvironment and its potential to treat retinal degenerative diseases [165,166,167]. The risks of carcinogenesis and immune rejection associated with mesenchymal stem cells are, at least in theory, mitigated by the EVs produced by these cells. In a mouse model of retinitis pigmentosa, Zhang et al. showed that MSC-EVs are taken up in all retinal layers after intravitreal injection, improving photoceptor structure and function and visual acuity [168].

The benefits of MSC-EV injection have been studied in the context of several disease processes. For example, Zhang et al. showed that MSC-EVs could be used to reduce hyperglycemia-induced retinal inflammation [169]. Seyedrazizadeh et al. showed that embryonic-stem-cell-derived EVs delivered via IV injection improved the survival of retina ganglion cells in the optic nerve crush model [170].

Similarly, in a rat model of chronic ocular hypertension, intravitreal injection of MSC-EVs reduced retinal damage, increased the number of retinal ganglion cells, and inhibited the activation of caspase-3, reinforcing the protective potential of MSC-EVs [171]. In rat models of glaucoma induced by microbead injection or laser photocoagulation, MSC-EVs prevented retinal nerve fiber layer degenerative thinning [172]. MSC-EVs have also shown potential in mitigating ischemia-reperfusion injury [173] and delaying the development of diabetic retinopathy [174].

Like MSC-EVs, human embryonic-stem-cell-derived EVs have been shown to alleviate retinal degeneration in RCS rats and promote retinal Muller cell retro-differentiation [175].

**Table 2 ijms-26-00836-t002:** Clinical trials and case reports of therapeutic ocular EVs. Data collected from “clinicaltrials.gov (accessed on 10 December 2024)”.

No.	Trial ID	Source of EVs	Phase	Disease	Administration	Status/Outcome
1	NCT06543667	Limbal stem cells	I	Dry eye syndrome	0.15 mL/single eye/one time, four times a day for 3 months	Unknown status
2	NCT03437759	Mesenchymal stem cells	I	Macular holes	50 μg or 20 μg of MSC-Exos in 20 μL of PBS dripped into the MH region during pars plana vitrectomy	50 µg of MSC-Exos resulted in anterior chamber inflammation. No inflammation with 20 µg of MSC-Exos. MSC-Exos application correlated with anatomical closure and BCVA improvement [176]
3	NCT04213248	Umbilical mesenchymal stem cells	I/II	Chronic graft versus host disease	UMSC-exo 10 ug/drop, four times a day for 14 days	Unknown status
4	NCT05738629	Pluripotent stem-cell-derived mesenchymal stem cell	I/II	Dry eye diseases post refractive surgery	Eye drops 0.125 mL/single eye/one time, four times a day for 12 weeks	Not yet recruiting
5	NCT05413148	Mesenchymal stem cells	II/III	Retinitis pigmentosa	Subtenon injection	Unknown status

Despite their potential in animal studies, human studies of ocular MSC-EVs are currently limited (Table 2). In a small pilot trial of seven patients with macular holes, MSC-EVs were shown to improve both anatomic and visual outcomes [176].

## 4. Ocular EV Safety

Perhaps the biggest current obstacle to the utility of vitreous sampling for diagnostic use is patient safety, such as the risk of endophthalmitis, intraocular inflammation, retinal detachment, and hemorrhage [177,178].

A large meta-analysis of 105,536 intravitreal anti-vascular endothelial growth factor (VEGF) injections between 2005 and 2009 found a rate of endophthalmitis of 0.049% [179]. In other studies during the early 2000s, the risk of endophthalmitis following intravitreal injection ranged from 0.019 to 0.077% [180,181,182,183,184]. In 2018, a retrospective cohort study of 818,558 anti-VEGF intravitreal injections found a rate of endophthalmitis ranging from 0.047 to 0.100% depending on the anti-VEGF agent used [185]. Notably, while the overall risk of endophthalmitis remains low after intravitreal injection, with similar rates between injections in the office compared to the operating room, rates of culture-positive endophthalmitis are significantly higher in the office setting [186].

RPE tears can occur, and rare events occur at increased rates following intravitreal anti-VEGF injections, especially in those patients with existing pigment epithelium detachments [187]. Therefore, pre-existing conditions must be considered when evaluating the risk–benefit ratio for a particular patient.

The rate of rhegmatogenous retinal detachment in patients receiving anti-VEGF intravitreal injections was found to be 0.013% per injection [188].

Elevation of intraocular pressure (IOP) is an often-transient event lasting a few hours after intravitreal injections, and IOP monitoring in patients receiving injections is recommended [189,190,191].

The risk–benefit ratio shows a reassuring trajectory, with new studies showing diagnostic utility, while developing technologies hold promise for safe and routine vitreous biopsy [192,193,194]. Standardized protocols, consideration of each patient’s pre-existing conditions, and individualized risk–benefit discussions will undoubtedly be part of any future EV-based treatments.

## 5. Conclusions

The use of intraocular EVs for diagnostic and therapeutic purposes is a promising research direction. The field of ocular EV research could draw from the rich experience of studying the circulating proteome in metabolic and cardiovascular heath, with the promise of discovering novel disease-related biomarker sets as well as the development of targeted assays [195,196]. One potential evolution in the study of ocular EVs is the isolation of cell-specific EVs, which was shown to be possible by Pulliam et al. and Sun et al. in the study of neuronal EVs [197,198,199]. Advances in the standardization of EV isolation, purification, and characterization along with safety and toxicity studies are needed to further enable clinical trials.

## 6. Future Directions

The field of diagnostic vitreous EV analysis is currently limited by sample availability, biasing studies to those disease processes where vitreous sampling is indicated (e.g., endophthalmitis) or is part of a medically necessary vitrectomy in the case of retina surgery. The development of safe and reliable routine vitreous sampling has the capacity to shed light on multiple other underexplored directions. Until vitreous sampling is as routine and safe as an intravitreal injection, study sample sizes will remain limited in the number and scope of diseases. Furthermore, numerous protocols and procedure modifications are currently used by researchers to study vitreous EVs, and standardization is needed to ensure the validity and translational potential of the results [200,201]. A specific focus on the isolation of ocular EVs from small biofluid volumes is essential, and slow development in this area likely contributes to the relatively slow pace of development of ocular EV-based biomarkers.

Therapeutic ocular EV applications are a promising field backed by robust animal research and experience from other fields of medicine. EV source cells could be chosen or modified to produce EVs with desirable characteristics, taking into account cell age and culture conditions. For example, EVs derived from IL-35-producing B-regulatory cells not only contained IL-35 but also suppressed neuroinflammation in a mouse autoimmune uveitis model [202]. Altering the EV cargo and surface peptides is another promising direction, albeit with added complexity in manufacturing and regulatory guidelines. For example, in a rat model of diabetic retinopathy, loading of small EVs with bevacizumab reduced the frequency of intravitreal injections [203]. Loading the MSC-EVs with pigment epithelium-derived factor (PEDF) inhibited endothelial cell proliferation and tube formation in an oxygen-induced retinopathy model in mice, opening the potential for EV cargo modification to further enhance the therapeutic effect [204]. New methods, such as the fusion of EVs with drug-loaded liposomes, are being explored [205]. Modifications of the EV phospholipid layer are also a promising way to alter the incorporation into retinal layers [206]. This field could draw heavily from the well-established research on liposomal drug formulation [207], as well as the ability to influence spatiotemporal EV release provided by encapsulation [136].

However, several challenges to ocular therapeutic implementation of EVs remain. First, the risks of intraocular injection, such as infection, inflammation, and elevations in intraocular pressure, must be balanced with the potential benefits. Second, the clinical benefit of therapeutic injection of a novel, complex clinical product must be justified compared to existing therapies. For example, the injection of encapsulated EVs carrying anti-VEGF may prolong the time between injections but introduce additional risks and costs due to the complexity of the clinical product.

Development of regulatory guidance is ongoing and urgently needed for safe and reproducible therapeutic EV application [208,209]. Researchers who aim to develop therapeutic EVs should pay special attention to the latest MISEV guidelines [210], with a specific focus on reproducible manufacturing, quantification, in vitro and in vivo functional assays, and, ultimately, therapeutic efficacy.

## Figures and Tables

**Figure 1 ijms-26-00836-f001:**
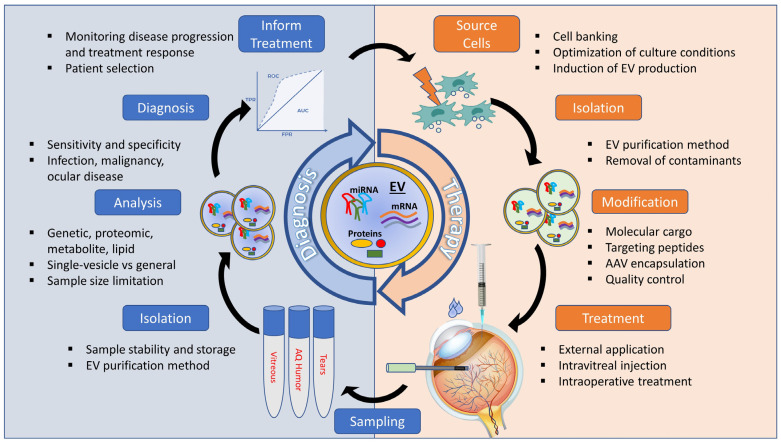
Schematic overview of the cycle of EV-based diagnosis and treatment in ocular eye disease.

**Figure 2 ijms-26-00836-f002:**
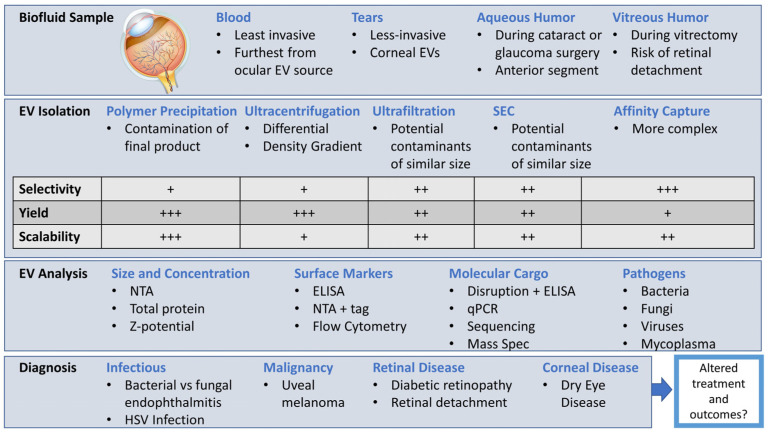
Overview of ocular EV isolation and analysis methods. +–+++ scale within table indicates relative ease for each EV isolation technique ranging from least to greatest respectively.

**Figure 3 ijms-26-00836-f003:**
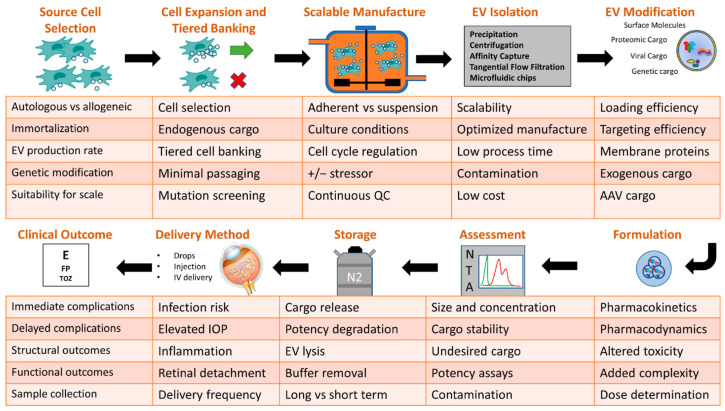
Key considerations for the manufacturing of therapeutic EVs for ocular use. Within the assessment “NTA” figure, green tracing represents a single group of particles with narrow size distribution. Red tracing represents a wide size distribution and a dimorphic particle population.

**Table 1 ijms-26-00836-t001:** Diagnostic clinical trials of EVs for ocular disease. Data collected from “clinicaltrials.gov (accessed on 10 December 2024)”.

No.	Trial ID	Source of EVs	Disease	EV Analysis	Status/Outcome
1	NCT06475027	Blood	Dry eye syndrome	miRNA	Not yet recruiting
2	NCT05888558	Serum	Ocular Myasthenia Gravis	miRNA	Enrolling by invitation
3	NCT03264976	Serum	Diabetic retinopathy	miRNA	Unknown status
4	NCT06198543	Plasma, atrial fluid, and vitreous fluid	Proliferative diabetic retinopathy	Proteomic analysis	Not yet recruiting
5	NCT06188013	Plasma	Diabetic retinopathy	Proteomic analysis	Not yet recruiting
6	NCT04164134	Blood	Retinoblastoma	RNA expression on platelets and allelic DNA balance of EVs in the blood of adult RB1 mutation carriers	Completed. No published results available.

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
