# Peer review of "Diagnostic and Therapeutic Utility of Extracellular Vesicles in Ocular Disease"

_ijms, 2025, doi:10.3390/ijms26020836_

Round 1
Reviewer 1 Report
Comments and Suggestions for Authors
The authors have submitted a detailed review article, which mainly summarized the diagnostic and therapeutic utility of extracellular vesicles (EV) in ocular disease. The EV isolation, quantification and analysis were introduced in Section 1. While the ocular diagnostic and therapeutic use of EV were reviewed in detail in Section 2 and Section3.
This review seems very meaningful. I think it can be accepted after minor revision.
1. The title of Figure 1 is incomplete.
2. Line 99, -served in the animal model [35].
3. In Section 3, the ocular therapeutic use of EV, what is the source of the EV? This part might be confusing, and I didn't get it, too. Did you discuss only ocular EVs, or all EVs used for ocular therapeutics (unlimited sources)? Please clarify this issue in your manuscript.
4. Section 3.4, please specify the mode of ocular administration of EV and the dosage form commonly studied. For example, eye injections or eye drops?
5. Section 5 and 6 can be integrated.
6. References, some references (more than 20) need additional information, such as page range. Please revise carefully.
Author Response
Comments 1: The title of Figure 1 is incomplete.
Response 1: Thank you for bringing this to our attention. The title of Figure 1 has been corrected accordingly.
Comments 2: Line 99, -served in the animal model [35].
Response 2: Thank you, the typo has been corrected.
Comments 3: In Section 3, the ocular therapeutic use of EV, what is the source of the EV? This part might be confusing, and I didn't get it, too. Did you discuss only ocular EVs, or all EVs used for ocular therapeutics (unlimited sources)? Please clarify this issue in your manuscript.
Response 3: Thank you for raising this concern, indeed the language use in the paper was broad and did not specify that the cellular source of EVs in ocular applications is for the most part not ocular cells, but rather MSCs. The wording in Sections 3 and 3.1 has been clarified to address this concern.
Comments 4: Section 3.4, please specify the mode of ocular administration of EV and the dosage form commonly studied. For example, eye injections or eye drops?
Response 4: Thank you for bringing this point to our attention. We have amended section 3.4 to specify that the commonly-studies form of dosing relies on delivering a specified amount of EV protein, and that further developments in the standardization of dosing are urgently needed.
Comments 5: Section 5 and 6 can be integrated.
Response 5: Thank you for the suggestion. We have combined Section 5 and 6 of the review into a single section.
Comments 6: References, some references (more than 20) need additional information, such as page range. Please revise carefully.
Response 6: Thank you for raising this concern. We have updated the reference style to follow the linked MDPI ACS style according to the journal guidelines, and page numbers are included in all instances where they are available by the publisher. All references also now include the doi as suggested by the journal, which will enable easy access to the cited publication by the reader.
Reviewer 2 Report
Comments and Suggestions for Authors
In this review, the authors summarized recent advances in the diagnostic and therapeutic use of ocular EVs. The authors also explored future directions by drawing on insights from the study of EVs in other organ systems and disease states. While this field holds significant potential for advancement, the review fails to comprehensively highlight recent breakthroughs and clinical applications. Thus, a major revision is recommended.
Specific Comments:
1. A more comprehensive summary of recent therapeutic cases utilizing EVs for ocular treatment is required. Incorporating additional tables and figures to better illustrate these advancements is encouraged
2. Please check the caption for Figure 1.
3. The section on the therapeutic administration of EVs is insufficient. Additional discussion is necessary, particularly regarding innovative delivery systems such as the use of microspheres for EV administration, which is missing.
4. The review should include a detailed summary of recent clinical trials and case studies on the diagnostic and therapeutic use of EVs in ocular diseases.
5. A thorough discussion of challenges associated with EV manufacturing, modification, isolation, purification, storage, and regulatory compliance is essential, especially for future clinical applications. The authors should include more recent references, such as 10.1016/j.tibtech.2024.08.007 doi.org/10.1038/s41565-021-00931-2, to highlight these challenges in the concluding section.
Author Response
Comments 1: A more comprehensive summary of recent therapeutic cases utilizing EVs for ocular treatment is required. Incorporating additional tables and figures to better illustrate these advancements is encouraged.
Response 1: We appreciate the reviewer’s comment and appreciate that a summary of ongoing clinical trials and case reports would be useful to the reader. A thorough search of clinicaltrials.gov as well as PubMed was conducted to identify case reports and clinical trials, demonstrating the scarcity of trials and an even more profound lack of results. The results of this search have been summarized in Tables 1 and 2.
Comments 2: Please check the caption for Figure 1.
Response 2: Thank you for bringing this to our attention. The title of Figure 1 has been corrected accordingly.
Comments 3: The section on the therapeutic administration of EVs is insufficient. Additional discussion is necessary, particularly regarding innovative delivery systems such as the use of microspheres for EV administration, which is missing.
Response 3: We thank you for your comment, and the suggestion to address novel and innovative EV delivery systems. Accordingly, the section on EV therapeutic administration has been expanded, especially focusing on EV manufacture, and modification, including the use of microspheres and hydrogel encapsulation.
Comments 4: The review should include a detailed summary of recent clinical trials and case studies on the diagnostic and therapeutic use of EVs in ocular diseases.
Response 4: Thank you for raising this concern. We have included two tables compiled from clinicaltrials.gov that include diagnostic and therapeutic use of ocular EVs. Unfortunately, the studies in humans are few, and just one study is reporting results.
Comments 5: A thorough discussion of challenges associated with EV manufacturing, modification, isolation, purification, storage, and regulatory compliance is essential, especially for future clinical applications. The authors should include more recent references, such as doi.org/10.1016/j.tibtech.2024.08.007 and doi.org/10.1038/s41565-021-00931-2, to highlight these challenges in the concluding section.
Response 5: We thank the reviewer for raising this concern. We have greatly expanded the corresponding section “3-EV Manufacture” to discuss EV manufacture with changes highlighted within the manuscript. While the topic of EV manucature is broad, with potentially hundreds of useful references, we have aimed for a balance of providing an overview of concepts and specific examples in the field of EV manufacture, while staying relevant to ocular disease. We have also expanded the conclusion section to include the suggested references.
Round 2
Reviewer 2 Report
Comments and Suggestions for Authors
The authors have only partially addressed my concerns. First, additional figures highlighting recent advancements in the application of therapeutic EVs for ocular treatment are necessary to strengthen the manuscript. While the supplemental discussion of EV clinical aspects is commendable, this section remains weak. Clinical applications and the translation of EVs, particularly their modification, should be discussed earlier than the manufacturing process, as modification techniques represent a primary focus to scalable manufacturing and a barrier for regulatory compliance. Issues such as contamination and heterogeneity introduced during modification can lead to genetical instability of the donor cells, side products, or the introduction of unwanted cargo in EV lumen. These challenges should be supported with more references and further discussion. Most importantly, the manuscript lacks a detailed examination of the significant barriers specific to the use of EVs for ocular treatment, rather than covering solely general challenges and obstacles.
Author Response
The authors have only partially addressed my concerns.
Comment 1: First, additional figures highlighting recent advancements in the application of therapeutic EVs for ocular treatment are necessary to strengthen the manuscript.
Response 1: Thank you for this comment. Accordingly, the existing Figure 1 has been modified to reflect the changes in the text and provide the reader with a quick overview of the topics covered in the review. Figures 2 and 3 have been added to reflect advances and key considerations in diagnostic and therapeutic EV applications respectively.
Comment 2: While the supplemental discussion of EV clinical aspects is commendable, this section remains weak. Clinical applications and the translation of EVs, particularly their modification, should be discussed earlier than the manufacturing process, as modification techniques represent a primary focus to scalable manufacturing and a barrier for regulatory compliance.
Response 2: The authors agree that the modification of EVs is a crucial part of the manufacturing process, and a large focus for achieving scalable manufacturing and regulatory compliance. We believe that the current organization of discussing EV production by cells, followed by modification is a logical way to organize the review. We discuss general considerations of EV modifications earlier in the review, specific examples of EV modification within their relevant sections.
We further added discussion points on regulatory compliance throughout the discussion of the manufacturing process and cited detailed review papers focusing solely on regulatory compliance in the field of therapeutic EVs.
Comment 3: Issues such as contamination and heterogeneity introduced during modification can lead to genetical instability of the donor cells, side products, or the introduction of unwanted cargo in EV lumen. These challenges should be supported with more references and further discussion.
Response 3: Thank you for this helpful comment comment. We have expanded the EV modification section with discussion related to heterogeneity, genetic instability, side products, and unwanted cargo.
Comment 4: Most importantly, the manuscript lacks a detailed examination of the significant barriers specific to the use of EVs for ocular treatment, rather than covering solely general challenges and obstacles.
Response 4: Thank you for this insightful comment, we agree that addressing significant barriers to EVs for ocular diagnosis and treatment should be a key portion of the discussion. General principles of EV manufacture, storage, and delivery apply equally to ocular and systemic applications, albeit with a difference in therapeutic volumes, but there are unique aspects related to the eye. Accordingly, we have added a paragraph to the “Future Directions” section of the review that highlight this topic. We have also commented on specific ocular challenges and barriers throughout the review.
Round 3
Reviewer 2 Report
Comments and Suggestions for Authors
The authors have addressed my concerns.